behaviour/ecology

central place foraging, colony size, Old World fruit bats, resources, movement, migration

**Author for correspondence:**
María C. Calderón-Capote
e-mail: mccalderonc@gmail.com

# Foraging movements are density-independent among straw-coloured fruit bats

María C. Calderón-Capote[1,2], Dina K. N. Dechmann[1,3], Jakob Fahr[1,4], Martin Wikelski[1,3], Roland Kays[5] and M. Teague O'Mara[1,3,6]

[1]Department of Migration, Max Planck Institute of Animal Behavior, Radolfzell, Germany
[2]Department of Biology, University of Munich, Munich, Germany
[3]Department of Biology, University of Konstanz, Konstanz, Germany
[4]TU Braunschweig, Zoological Institute, Braunschweig, Germany
[5]North Carolina Museum of Natural Sciences and Department of Forestry and Environmental Resources, North Carolina State University, Raleigh, NC, USA
[6]Department of Biological Sciences, Southeastern Louisiana University, Hammond, LA, USA

MCC-C, 0000-0002-7646-3082; DKND, 0000-0003-0043-8267; JF, 0000-0002-9174-1204; MW, 0000-0002-9790-7025; RK, 0000-0002-2947-6665; MTO, 0000-0002-6951-1648

Intraspecific competition in large aggregations of animals should generate density-dependent effects on foraging patterns. To test how large differences in colony size affect foraging movements, we tracked seasonal movements of the African straw-coloured fruit bat (*Eidolon helvum*) from four colonies that range from 4000 up to 10 million animals. Contrary to initial predictions, we found that mean distance flown per night (9–99 km), number of nightly foraging sites (2–3) and foraging and commuting times were largely independent of colony size. Bats showed classic central-place foraging and typically returned to the same day roost each night. However, roost switching was evident among individuals in three of the four colonies especially towards the onset of migration. The relatively consistent foraging patterns across seasons and colonies indicate that these bats seek out roosts close to highly productive landscapes. Once foraging effort starts to increase due to local resource depletion they migrate to landscapes with seasonally increasing resources. This minimizes high intraspecific competition and may help to explain why long-distance migration, otherwise rare in bats, evolved in this highly gregarious species.

# 1. Introduction

Foraging behaviour varies with the number and behaviour of competitors [1,2]. Living near large numbers of conspecifics can

incur both costs and benefits. On the one hand, large numbers of individuals in a group can reduce predation risk and improve information sharing about the quality and availability of resources [3–6]. For instance, central place foragers such as polydomous leaf-cutting ant species form large connected nests and may benefit from these extensive colonies by increasing foraging success on both clumped and dispersed resources [7,8]. They can also gain information about profitable resources at the central place by reducing time spent at the foraging sites [9]. They thus maximize rate of delivery and food processing inside their non-mobile nests [10]. On the other hand, large group size can also be costly for individuals if it increases intraspecific competition [11]. Optimal foraging theory predicts that animals should maximize food intake while minimizing the mean time spent foraging, energy expenditure and travel distance to the food patches [12,13]. For central place foragers, intraspecific competition generates additional density-dependent effects on foraging patterns and individual fitness due to the constraint of returning to the roost [11,14]. With higher numbers of individuals, resources near the central place are depleted faster, forcing animals to make longer foraging trips. This is true for colonial seabirds where suitable breeding locations are limited [15–17]. Larger colonies of breeding pelicans (*Pelecanus occidentalis*) have greater foraging areas due to local intraspecific competition and reduced prey availability, which may also trigger migratory movements [15]. A comparison between two colonies of cape gannets (*Morus capensis*) revealed larger, longer and faster foraging trips in the bigger colony, as well as double the number of foraging sites [16]. Similarly, penguins (*Pygoscelis adeliae*) increase foraging distance to cope with reduction of food resources around their roosts despite reaching energetic thresholds [17]. Although there is a clear influence of colony size on the foraging behaviour of colonial animals, no study has addressed density-dependent effects in bats where colony size sometimes scales over several orders of magnitude. Here, we examined the African straw-coloured fruit bat, *Eidolon helvum*, a central place forager, which seasonally migrates causing large fluctuations in colony size. Colonies can vary from 100–100 000 individuals, to one of the largest vertebrate aggregations estimated to comprise 5–10 million individuals during a three-month period per year at Kasanka National Park (NP) in Zambia.

At least in part due to these large aggregations, *E. helvum* provide important ecosystem services as they fly across landscapes and transport seeds and pollen over larger distances than recorded for any other vertebrate [18,19]. Seasonal migration results in some colonies growing to several million bats for short periods, while other colonies are seasonally abandoned, and still others appear to maintain a low baseline resident population [20–22]. Previous work suggested that foraging movements of *E. helvum* vary with colony size [20]. When the colony in Accra, Ghana, is at peak size during the local dry season (*ca* 150 000), bats fly a mean distance of about 88 km one way to access food [20]. However, during the wet season when only 4000–5000 resident bats remain, they forage over much shorter distances with a mean of 37 km. This 53% increase suggests a strong density-dependent effect on foraging distance. However, the change to longer foraging distance coincides with a seasonal shift to nectar resources suggesting that there may be additional effects at this site. To gain a more general understanding of how population density influences the foraging behaviour of these bats we then need to examine additional sites across seasons and population sizes.

We tracked individuals in the largest known colony of these bats in the Kasanka NP, Zambia to explore potential density-dependent effects on foraging behaviour. An estimated 5–10 million *E. helvum* aggregate there during the local wet season from late October to December [21,22]. We compared these movement data with published data from three other colonies of seasonally variable size in West Africa. We predicted that (i) foraging distance will increase proportionally to colony size; and (ii) as individuals travel longer distances the increased time spent commuting will result in reduced foraging time. We also predicted that (iii) if colony size increases as a function of food availability, bats in a large colony should follow marginal value theorem predictions [13,14], spend more time at each feeding site, and therefore visit fewer sites during each night. Individuals should leave a patch when the benefits of foraging are equal to those of the surrounding environment. They should then move to another patch to enhance food acquisition. Finally, if depleted resources force bats to migrate away from an area, we predicted that (iv) the number of nightly foraging sites should increase during the time of colony size decline, shortly before the onset of migration. If individuals explore more foraging sites during low resource density, we expected them to reduce effort invested in foraging by choosing alternative roosts, reducing central place foraging. Our results will help understand foraging decisions in colonial species as well as contribute to our understanding of the drivers and triggers of the long-distance migration of this ecological keystone species across Africa.

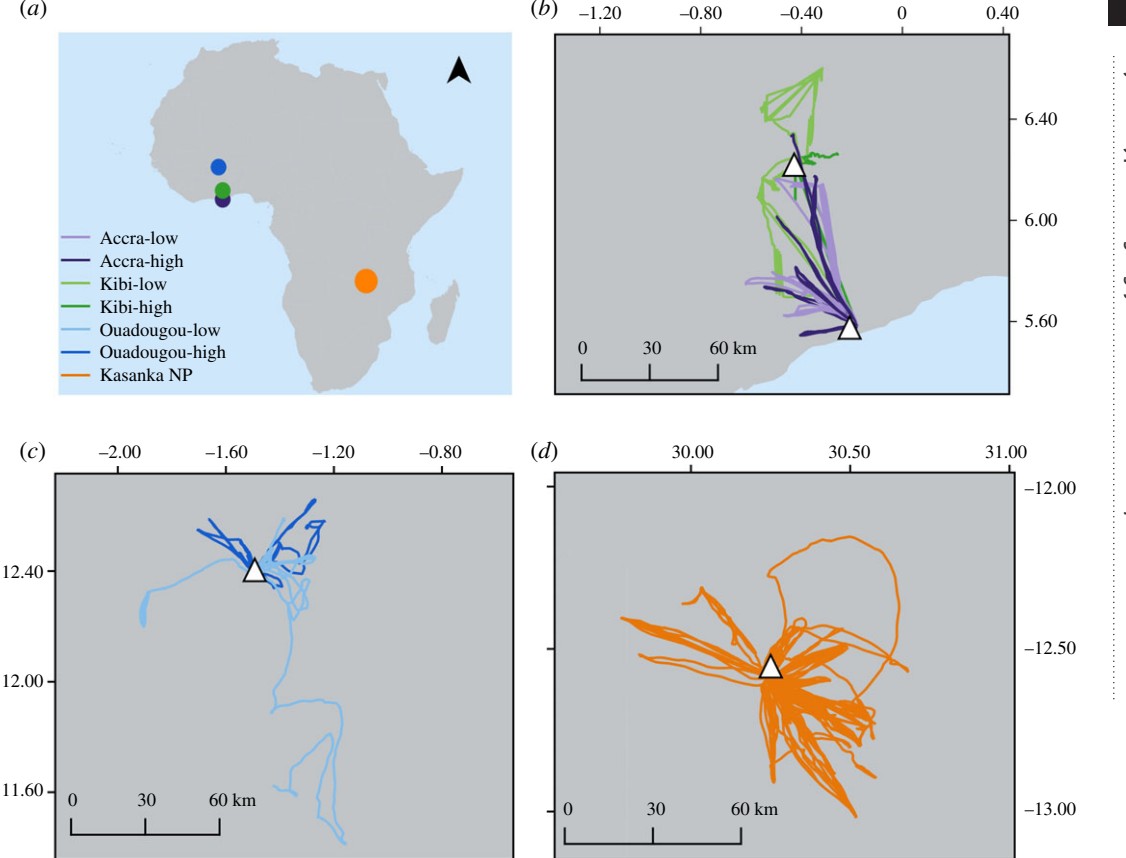

**Figure 1.** Foraging tracks and movement patterns of *Eidolon helvum* (*n* = 45) in Africa. (*a*) Map of Africa indicating the location of each colony; (*b–d*) Tracks of bats in Accra and Kibi (*b*), Ouagadougou (*c*) and Kasanka (*d*). White triangles in the foraging tracks represent the main roosting colonies. [Colony name]-low, low population size; [Colony name]-high, high population size.

## 2. Methods

### 2.1. Description of *Eidolon helvum* colonies

Straw-coloured fruit bats occur in large mixed-sex colonies year-round across the tropical belt of Africa [21]. They migrate seasonally as far north as the Savanna zone and as far south as the Cape Province in South Africa [23,24]. We compared data from the following four colonies (figure 1*a*).

#### 2.1.1. Accra, Ghana

This colony resides in the gardens of the 37 Military Hospital in Accra (5°35′ N, 0°11′ W). It persists throughout the year, but varies seasonally in size. Our data are from individuals tracked during the dry season (Accra-high; February, *n* = 9) when the colony was at its maximum size (150 000 bats) and during the wet season when colony size was low (Accra-low; 4000 bats; August; *n* = 8; figure 1*b*; electronic supplementary material, table S1). Most bats in this colony migrate northwards with the onset of the wet season in March/April [25]. Precipitation in this region is bimodal, but with little temperature variation.

#### 2.1.2. Kibi–Old Tafo, Ghana

This smaller colony roosts in the garden of the king's palace in Kibi (6°15′ N, 0°24′ E) about 76 km from Accra. It also varies seasonally in size but persists throughout the year. Individuals from Kibi frequently switch to a roost in Old Tafo approximately 17 km southwest. We consider this as one single interacting colony, and refer to it as Kibi. Here, we used data from bats tracked during the wet season's low colony size (Kibi-low; 4000 bats; August–September; *n* = 8) and during the dry season in January when colony

size is high (Kibi-high; 41 000 bats; $n = 4$; figure 1b; electronic supplementary material, table S1). Climate is bimodal as in Accra, but precipitation is higher (up to 1600 mm) [18].

### 2.1.3. Ouagadougou, Burkina Faso

The colony is located in the Parc Urbain Bangr-Weogo inside Ouagadougou (12°24′ N, 1°30′ W). This colony is present from February to September and completely absent during the rest of the year. Data included individuals tracked when the colony was at peak size (Ouagadougou-high; 67 000 bats; June–July; $n = 7$), and when colony size was decreasing shortly before migration onset (Ouagadougou-low; 38 000 bats; August–September; $n = 5$; figure 1c; electronic supplementary material, table S1). Ouagadougou is situated in the Sahel zone [26]. Its humid tropical climate is characterized by a single dry season from November until March, and a wet season between April and October [27]. Mean annual rainfall is 782 mm. Fructification peaks occur in the wet season [27], whereas in the dry season vegetation is scarce. Note that for the Ouagadougou colony the two sizes occur within the same season.

### 2.1.4. Kasanka NP, Zambia

This colony roosts in a small patch of swamp fig (mushitu) forest (12°34′ S, 30°11′ N) within Kasanka NP. It is a migration stop-over or endpoint and occurs only during October–January each year. It is estimated to comprise about 5–10 million individuals at peak size. We used data from 20 individuals tracked during the wet season in November–December of 2013 and 2014 (figure 1d; electronic supplementary material, table S1). The colony arrives just before the onset of rains and includes the first part of the rainy season [22]. The colony is composed of individuals that probably migrate from various colonies in the Congo River Basin [21,22].

## 2.2. Logger settings and data collection schedule

We obtained a mix of new and previously published tracking data (available at the Movebank Data Repository (doi:10.5441/001/1.k8n02jn8), Scharf et al. [28]). Data were collected in 2009–2014 from male E. helvum (278.4 ± 20.5 g) equipped with GPS loggers (e-obs, GmbH, Munich, Germany). Mean total logger mass was 20.95 ± 0.90 g, which was 7.56 ± 0.59% of body mass. Females were not tracked as loggers exceeded recommended weight limitations for these smaller animals. The loggers collected GPS locations, flight speed and altitude, as well as triaxial acceleration data. Loggers were programmed according to different regimes (due to technical development of the loggers) henceforth called 'cohorts' and attached to the individuals by either gluing or using collars (see Fahr et al. [20] and Abedi-Lartey et al. [18] for details; also electronic supplementary material, table S1). Cohort 1 collected GPS fixes at a regular interval of 600 s during GPS on-times. Cohort 2 collected GPS fixes every 900 s until the animal was moving at a speed of greater than or equal to $5 \, \mathrm{ms}^{-1}$, after which they switched to an interval of 300 s [20]. GPS fixes for Cohort 3 and Cohort 4 were acceleration-informed (see Brown et al. [29]). They collected fixes every 1800 s until the bat started flying, then switched to fixes every 300 s (Cohort 3) and 150 s (Cohort 4; electronic supplementary material, table S1). Cohort 3 and Cohort 4 started data collection immediately at release of the animal. GPS-on times were 16.00/18.00–06.00 local time for all cohorts. Acceleration data were collected for all cohorts using a 14.09 s-interval each minute with a frequency of 18.74 Hz, except in Kasanka during 2014 where a 13.4 s-interval per minute with a frequency of 20 Hz was used. Loggers on bats from Accra-low used GPS settings from Cohort 1 and 2, Accra-high from Cohort 3, Kibi from Cohort 3 and 4 and Ouagadougou and Kasanka from Cohort 4.

## 2.3. Behavioural classification

We used only data from individuals with at least one complete night of tracking, on average: 4.4 ± 1.9 nights (see electronic supplementary material, table S2). We classified the GPS locations into three behaviours (resting, commuting and foraging) before analysing foraging parameters, based on the percentage of flapping flight identified within an accelerometry segment [30] (additional description, see moveACC: https://gitlab.com/anneks/moveACC). The percentage of flapping bouts per GPS fix was used to determine the behavioural classes. 'Resting' included all locations with a percentage of flapping lower than 10%; 'foraging' included all locations with a percentage of flapping higher than 10% but lower than 50%, which indicated active movement in a foraging site but not commuting; and

'commuting' included all locations with a percentage of flapping higher than 50% that indicated bats were commuting from roosts to foraging sites or between foraging sites. Additionally, we distinguished roosting sites as the GPS locations used before sunset and after sunrise each day.

## 2.4. Movement analysis

We calculated total distances moved from the colony to the foraging sites per night as the sum of the straight-line segment distance between successive GPS points, from the first point in the evening before a bat left the roost to the last point at the beginning of the next day when it had arrived at the roost. The maximum foraging distance was calculated as the Euclidian distance to the furthest point from the colony that was classified as foraging.

We used first passage time analyses (FPT) in *adehabitatLT* [31] to identify nightly foraging sites in the four colonies across seasons. FPT is defined as the time an animal takes to pass a given radius, which can be interpreted as the animal's search effort and indicates the areas where individuals interact with the landscape. FPT is scale-dependent, therefore we calculated the FPT at every location along the track of each bat for a radius of 1–100 m by 1 m increments. Variance plots of the FPT showed individual peaks around 10 and 60 m, which were consistent among all individuals, and we used a 60 m radius to distinguish foraging sites (electronic supplementary material, figure S1a). The variance peaks correspond to the spatial scales that best differentiate between high and low passage times. We used posterior Lavielle segmentation [32,33] to identify the places where each animal spent substantial periods of time of 500–4000 s to exclude locations where animals did not allocate foraging effort. This is a penalized contrast function that allows for an indeterminate number of change points [32], and we graphically chose the number of segments in each trajectory based on the decrease of the contrast function. When a clear break was observed, we assigned this as the number of probable segments (electronic supplementary material, figure S1b) within each individual trajectory. After the segmentation of the tracks (electronic supplementary material, figure S1c), segments that corresponded to foraging sites were inspected in QGIS [34] contrasting them with previous behavioural classification of the GPS locations.

## 2.5. Activity budgets

After the behavioural classification, we calculated activity budgets for each colony as the proportion of time each bat spent commuting, foraging and resting during the night excluding the time bats spent roosting each day.

## 2.6. Day roost fidelity in a central place forager

We describe day roost fidelity in the different colonies to identify modifications of central place foraging in *E. helvum*. This was done by examining each individual track. Day roost fidelity was quantified as the proportion of all nights each bat returned to a given roost. A day roost was defined as the location where the last morning and first evening GPS fix overlapped for one or more individuals in one or more days [35].

## 2.7. Statistical analyses

All analyses were completed in R 3.5.1 [36]. We used generalized linear mixed effects models (glmm) in *MASS::glmmPQL* [37] to describe differences in movement patterns. Specifically, we assessed the influence of colony size on the foraging parameters total distance, maximum foraging distance and number of foraging sites. Model fits were evaluated by their $R^2$ value in *r2glmm* [38]. We then modelled distance variables with a gamma distribution given the right-skewed nature of our data and the number of foraging sites with a Poisson distribution as appropriate for count data. All glmm included animal identity as a random effect. We also performed multiple comparisons with Tukey's test using the *multcomp* package [39] to evaluate the differences among colonies. We also test whether differences in maximum total distances (electronic supplementary material, figure S2) were larger in bigger colonies with Kruskal–Wallis test. In addition, we used glmm in *lme4* [40] to tested the effect of travel distances on the number of foraging sites. To test for differences in activity budgets among colonies, we used Kruskal–Wallis tests to compare the amount of time spent commuting, foraging and resting at night. We also tested for colony effects on commuting distance and commuting time using a

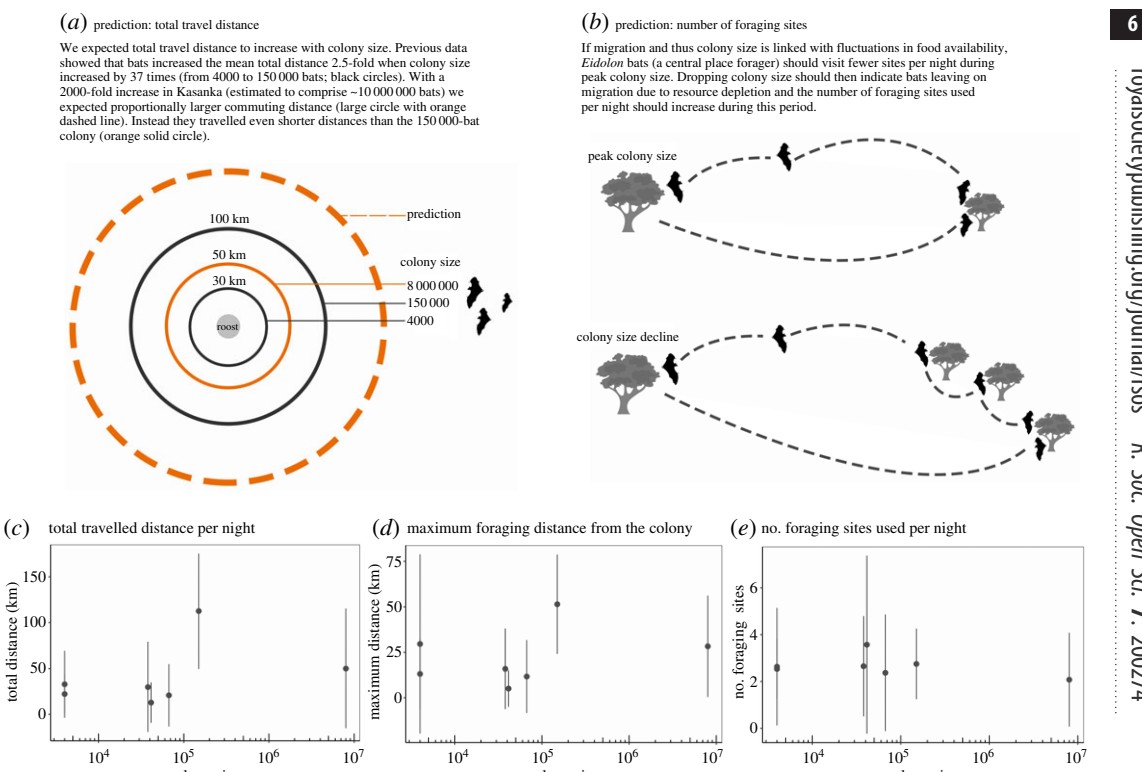

**Figure 2.** Foraging movement patterns of *Eidolon helvum* (*n* = 45). Predictions of (*a*) total distance travelled, (*b*) the number of foraging sites. Observed foraging patterns that do not follow initial predictions: (*c*) total distance per night across all colonies, (*d*) maximum distance to the furthest foraging site across all colonies, and (*e*) number of foraging sites used per night across all colonies. For better representation, colony size on the *x*-axis is shown as logarithm to the base 10. In panels (*c*–*e*), results for each colony are always shown in the following order according to Accra- and Kibi-low (4000), Ouagadougou-low (38 000), Kibi-high (41 000), Ouagadougou-high (67 000), Accra-high (150 000), Kasanka (8 000 000).

likelihood ratio test of nested models. The more complex model included commuting distance as response variable, commuting time and colony as fixed effects, and individuals as a random factor; and the second simpler model excluded colony as fixed effect. Finally, we used Kruskal–Wallis tests to examine differences in day roost fidelity among colonies. Whenever we found significant differences in the Kruskal–Wallis tests, we performed multiple comparisons between colonies with the Dunn test due to unequal sampling in each colony. Significance level was $p < 0.05$, and results are reported as means ± s.d.

# 3. Results

## 3.1. Movement patterns

The glmms showed that bat foraging behaviour was similar among the colonies, with the clear exception of Accra-high (figure 2). Increase in colony size in *E. helvum* was not reflected in the mean total distance flown per night (figure 2*c*). Against our expectations, bats in Kasanka travelled shorter mean total distances (48.6 ± 17.7 km; range 5.9–97.7 km) than those in the smaller Accra-high (99.8 ± 17 km; range 91.0–138.0 km), despite the 53 times larger population size. When we compared maximum total distances flown per individual, it was again found that bats in Accra-high covered higher maximum total distances (Kruskal–Wallis = 18.11, d.f. = 6, $p = 0.005$, electronic supplementary material, figure S2), while all other colonies were similar. Maximum foraging distances were relatively constant (mean range: 17.0–35.9 km; figure 2*d* and table 1), again except for Accra-high (49.3 ± 16.2 km; range 31.8–70.3 km) and thus not driven by colony size. Multiple comparisons with the Tukey's test confirmed that only Accra-high differed from the smaller colonies, but was similar to the Kasanka megacolony in maximum foraging distance (table 1). The number of foraging sites used per night did not differ between colonies either, and varied between 2.1 and 3.6 sites (figure 2*e* and table 1). The only

**Table 1.** Generalized linear mixed model predictors for foraging patterns. Significance terms: ***0.001, **0.01, *0.05. AC-low, Accra-low; K-low, Kibi-low; OU-low, Ouagadougou-low; K-high, Kibi-high; OU-high, Ouagadougou-high; AC-high, Accra-high; KA, Kasanka. Different superscript letters indicate significance between colonies using multiple comparison with Tukey's test. Colony groups that do not share a letter were different from each other. Symbols $\bar{x}$: mean; $\sigma$, standard deviation.

| seasonal colonies | colony size | total travelled distance ($\bar{x} \pm \sigma$) | total travelled distance ~ colony intercept | total travelled distance ~ colony estimates | maximum foraging distance ($\bar{x} \pm \sigma$) | maximum foraging distance ~ colony intercept | maximum foraging distance ~ colony estimates | foraging sites ($\bar{x} \pm \sigma$) | foraging sites ~ colony intercept | foraging sites ~ colony estimates |
|---|---|---|---|---|---|---|---|---|---|---|
| AC-low n = 6 | 4000 | 32.8 ± 3.9 | 29.2[ab] | | 27.5 ± 25.2 | 17.4[ab] | | 2.5 ± 0.6 | 0.9[ab] | |
| K-low n = 5 | 4000 | 20.3 ± 5.8 | | −9.5[ab] | 17.0 ± 12.4 | | −4.6[ab] | 2.6 ± 1.2 | | 0.04[ab] |
| OU-low n = 5 | 38 000 | 31.9 ± 12.8 | | −0.8[a] | 26.8 ± 7.5 | | −2.3[ab] | 2.7 ± 1.1 | | 0.04[ab] |
| K-high n = 3 | 41 000 | 13.0 ± 8.8 | | −16.6[ab] | 8.8 ± 5.8 | | −12.3[*a] | 3.6 ± 1.9 | | 0.3[b] |
| OU-high n = 6 | 67 000 | 22.9 ± 14.2 | | −8.4[a] | 22.3 ± 13.8 | | −6.4[a] | 2.4 ± 1.2 | | 0.05[ab] |
| AC-high n = 4 | 150 000 | 99.8 ± 17.0 | | 77.7[***c] | 49.2 ± 16.2 | | 31.2[***c] | 2.8 ± 0.8 | | 0.1[ab] |
| KA n = 16 | 8 000 000 | 48.6 ± 17.7 | | 15.0[b] | 35.9 ± 14.8 | | 7.7[bc] | 2.1 ± 1.0 | | 0.2[a] |
| $R^2$ | | | | 0.55 | | | 0.41 | | | 0.08 |

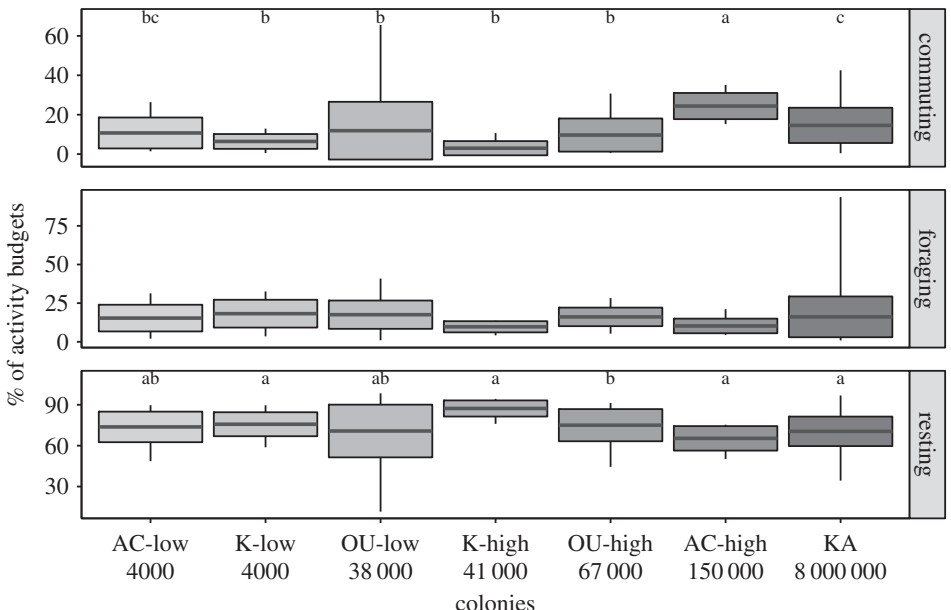

**Figure 3.** Activity budgets shown as a percentage of the total night activity per bat and colony. AC-low, Accra-low; K-low, Kibi-low; OU-low, Ouagadougou-low; K-high, Kibi-high; OU-high, Ouagadougou-high; AC-high, Accra-high; KA, Kasanka. Numbers on the x-axis refer to the size of each colony. Letters indicate differences among colony groups based on the Dunn test.

difference we observed in the number of sites used was between Kasanka (2.1 ± 1.0) and Kibi-high (3.6 ± 1.9) that showed the lowest and highest mean number of foraging sites (figure 2e). Moreover, bats that travelled farther did not use more feeding sites ($F_{1,196}$: 2.84, d.f. = 1, $p > 0.1$; electronic supplementary material, figure S3 and table S3).

## 3.2. Behavioural activity budgets

Bats spent 3–24% of their activity budgets commuting, which translated into 19–160 min. We found that commuting time differed among colonies (Kruskal–Wallis = 54.69, d.f. = 6, $p < 0.001$), but these differences were independent of colony size. Instead they were linked to longer foraging distances in Accra-high resulting in longer commuting time compared to the rest of the colonies (figure 3), as well as the considerably longer commuting time of bats in the largest colony, Kasanka, compared to bats in Kibi-high and Kibi-low (Dunn's test: $p < 0.001$). Based on these results, it was surprising that time spent foraging was similar in all colonies. Regardless of how much time they spent commuting, individuals spent 10–18% of their time (60–88 min) foraging every night (Kruskal–Wallis = 7.85, d.f. = 6, $p = 0.25$, figure 3). Additionally, time spent resting at night varied from 65 to 87% (295–516 min), but this was not influenced by colony size (Kruskal–Wallis = 28.17, d.f. = 6, $p < 0.001$). Moreover, exploration of bat commuting activity showed a significant relationship of the number of hours spent commuting and the commuting distance. Bats reached a foraging site at a similar speed in all colonies, as it is evidenced by their similar slopes, despite the different commuting distance in each colony (figure 4 and table 2).

## 3.3. Day roost fidelity

We observed substantial variation in the proportion of days bats roosted away from the main day roost (Kruskal–Wallis = 15.38, d.f. = 6, $p < 0.05$; electronic supplementary material, figure S4a), but only Accra-low and Ouagadougou-low were significantly different (Dunn test $p$-value = 0.04; electronic supplementary material, figure S4a). Day roost fidelity was highest in Accra where no bat used alternative day roosts regardless of population size. Day roost fidelity was variable at the nearby Kibi colony, where bats both at Kibi-low and Kibi-high frequently moved day roosts (electronic supplementary material, figure S4a). In Ouagadougou-low, bats also switched day roosts regularly (electronic supplementary material, figure S4a). All five individuals used alternative day roosts and one individual roosted always away from the main day roost, and never came back to the main roost

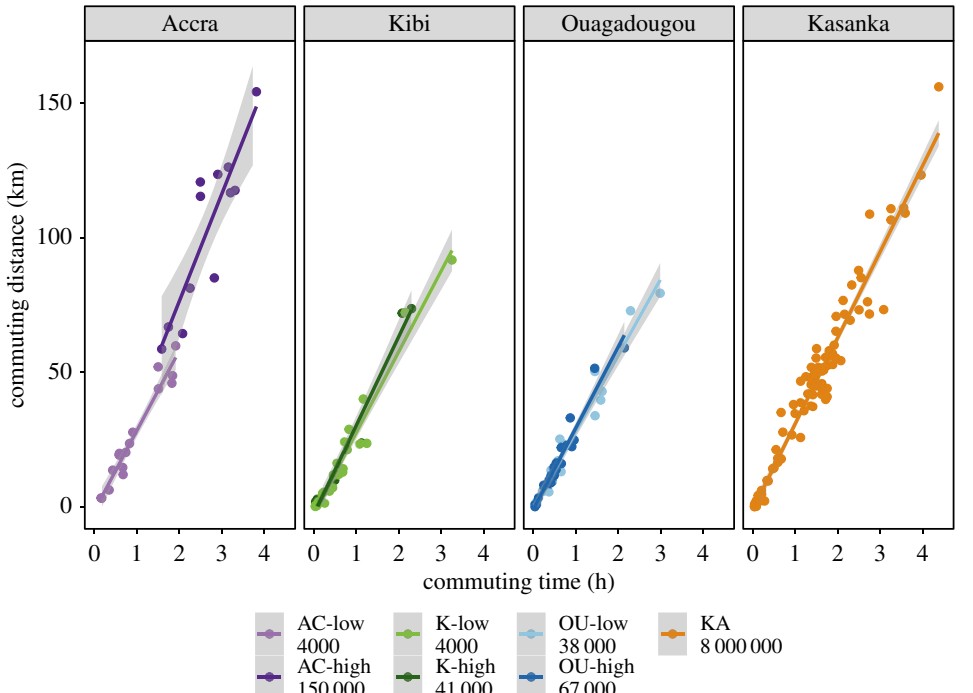

**Figure 4.** Commuting activity across colonies indicating the distance and the time commuting per night. Shaded areas in the predicted lines represent the 95% confidence interval. Abbreviations as in figure 3.

**Table 2.** Generalized linear model predictors for commuting activity. Significance terms: ***0.001, **0.01, *0.05. CI, 95% confidence interval.

| | commuting distance ~ time * colony | | commuting distance ~ time | |
| --- | --- | --- | --- | --- |
| | estimate | CI | estimate | CI |
| intercept | −1.2 | −7.4, 5.2 | −1.4 | −3.5, 0.6 |
| Accra-low/estimate | 29.5*** | 23.9, 35.2 | 32.2*** | 31.0, 33.3 |
| Kibi-low | −2.1 | −7.0, 11.3 | | |
| Ouagadougou-low | 0.2 | −7.6, 7.9 | | |
| Kibi-high | 0.8 | −7.9, 9.7 | | |
| Ouagadougou-high | −0.05 | −7.8, 7.7 | | |
| Accra-high | 10.5 | −4.6, 25.7 | | |
| Kasanka | 0.4 | −6.4, 7.2 | | |
| time: Kibi-low | −7.7 | −19.0, 3.5 | | |
| time: Ouagadougou-low | −0.5 | −7.3, 6.3 | | |
| time: Kibi-high | −8.08 | −20.9, 4.7 | | |
| time: Ouagadougou-high | 0.9 | −6.7, 8.6 | | |
| time: Accra-high | 4.3 | −3.3, 11.9 | | |
| time: Kasanka | 2.0 | −3.8, 7.8 | | |
| observations | | | | 184 |
| AIC | | 1189.03 | | 1251.94 |

during the four nights of tracking (electronic supplementary material, figure S4b). In Ouagadougou-high, the mean proportion of roost fidelity was higher than in Ouagadougou-low, and one bat used alternative day roosts during the whole tracking period (electronic supplementary material, figure S4c). Finally, the

Kasanka colony exhibited a high proportion of roost fidelity (electronic supplementary material, figure S4d). Six bats had two different day roosts except for one bat that had four different day roosts including the main roost (for practical reasons only bat 4156 is shown in electronic supplementary material, figure S4d). During these occasions, the bats remained in the vicinity of the foraging area until the next day when they returned to the main roost.

# 4. Discussion

In contrast to our expectations, we found few effects of massive seasonal and geographical variation in colony size on the foraging behaviour of *Eidolon helvum* at four sites across Africa. Unlike colonial sea birds [5,15,41], there was no effect of colony size on the daily distances travelled, duration of foraging, or the number of foraging sites used. We had expected higher values for total distance and maximum foraging distance (see orange dashed circle in figure 2a) in the Kasanka colony based on previously published foraging distances of *E. helvum* [20]. These values should have been vast, particularly for foraging distance, and especially in comparison to small colonies such as Accra-low and Kibi-low. However, these expectations were not met. Although the two largest colonies did have the longest maximum commuting times (1.2 times larger for Accra-high and 2.2 times larger for Kasanka) and maximum total distances flown (2.5 times larger), this increase did not match the magnitude of their 38–2000-fold increase in size compared to the smaller colonies (figure 2c–e; electronic supplementary material, figure S2).

We suggest that this is a result of synchronous seasonal fruiting events that attract migratory bats to places like Kasanka in the first place [22,42]. The fact that individuals living in a colony with millions of bats have similar foraging distances to bats in smaller colonies strongly suggests that food abundance is not a limiting factor for any of these populations. Bats at these sites also showed similar numbers of feeding bouts and total feeding time, suggesting that bats restrict their foraging activity to known and profitable food patches. This is supported by our observation that bats tended to exploit the same foraging patches repeatedly across nights, even at the highest population size. Food availability is directly tied to any aspect of density-related changes in foraging behaviour and central place foraging forces animals to maximize energy intake per unit of time [43,44].

It is interesting that the two largest colonies showed similar maximum flight distances in a single night (165 km). These long distances may be a result of slightly increased feeding competition, which might also show that these bats reached a limit to their total travelled distance per night. Their total possible flight time that maintains a positive energy balance if they were to return to their central day roost while meeting daily energetic needs could have been reached, similar to other tropical frugivorous bats [45]. However, in Accra-high, these longer distances were accompanied by decreased foraging times. Generally, the largest outlier in most measures of foraging was Accra-high. However, this could also be associated with a change in diet. During the dry season, *E. helvum* in Accra foraged extensively on flowers, mainly of kapok trees (*Ceiba pentandra*), despite the availability of many of the types of fruits that they feed on during the wet season [20]. It is possible that feeding on the nectar and pollen of these flowers has a higher overall energetic return rate than fruit [46].

At all other sites and seasons besides Accra-high, *E. helvum* generally feed on fruits [20,22,47]. This is supported by the observation that bats in Accra switched foraging sites more frequently during the flowering season when colony size was high than in the fruiting season when colony size was low. The Accra and Kibi sites in Ghana are only separated by *ca* 80 km (a feasible foraging distance for these bats) and during the period of peak population size in the dry season one would expect both populations to forage for similar food items. However, despite their close proximity the landscapes around Accra and Kibi differ dramatically, and the emphasis on flowers only in the Accra colony during this season makes the comparison between these sites particularly interesting for future work. Bats in Accra-high travelled a mean total distance of $99 \pm 17$ km while bats in the smaller Kibi-high flew only $13 \pm 9$ km. This seems to indicate that there are indeed compounding density effects on foraging when larger aggregations of bats occur locally in southern Ghana, although a larger sample size including ground-truthing of food choice will be necessary to confirm this. Furthermore, in Ouagadougou, Burkina Faso, decreasing food availability may be linked to decreasing population size (as a consequence of bats beginning to migrate away) and increased total foraging distance. It has been postulated that fruit bats migrate into Savanna zones, such as the one surrounding Ouagadougou, to profit from high food availability and low competition with resident species during the brief local wet season [42]. As we tracked bats at the tail end of their presence in Ouagadougou, it

is likely that food availability was already declining and thus bats possibly had to switch foraging sites more often towards the end of the wet season when bats begin to abandon the colony [47].

Long-distance migration in bats is a moderately rare phenomenon, particularly for fruit bats [48,49]. Migration in *E. helvum* seems to be a direct response to seasonal shifts in fruit availability across Africa [42], and it may be that this migration pattern evolved in response to the relatively stereotyped foraging behaviour we found across our sampled colonies. For example, pelicans are more likely to migrate if they roost in large colonies near marginal foraging grounds, suggesting that competition for resources pushes these birds to new sites [15]. If the return rate of a patch is low relative to the landscape around it, an optimal forager should quit the patch and move on to the next [14,50], and could then result in the continental migration inferred for *E. helvum* [42]. However, this could imply broad knowledge about the surrounding resource landscape, and in the case of migratory species, may integrate longitudinal experience of what the landscape (or next site) should be relative to current conditions. The large aggregations that *E. helvum* prefer could be powerful collective sensing systems that can detect shifts in the resource landscapes [51–53]. Relatively few individuals with decreased foraging efficiency could trigger the movement of these large colonies to their next locations [54], meaning that each individual does not need to make a migration decision based on their own immediate foraging returns alone. This would lead to the consistent foraging patterns that we found across these diverse sites. Similar to green wave surfing by migrating ungulates [55], *E. helvum* populations are likely to depend on massive increases in food availability across Africa as a consequence of seasonal precipitation [20,25,42].

Our results offer insight into the remarkable consistency of foraging patterns of a key ecosystem service provider with a wide distribution range across Africa [19]. We were unable to control for all environmental factors including difference in habitat and, most importantly, resource availability. Foraging behaviour of *E. helvum* was not directly linked to the size of the colony across sites, but there may be density effects across seasons within a site that require additional data to clarify. Our main result is that *E. helvum* maintained foraging distances and times even when population sizes were extreme. The migratory behaviour of these bats allows them to use central place foraging in spite of the tendency to form the large or even gigantic colonies they obviously prefer. This also suggests that despite the broadly accepted view that density-dependent competition is the norm in colonial species, straw-coloured fruit bats are a special case, as they can temporarily form a colony of several million individuals with few evident effects of competition that are probably mitigated through high availability of a broad range of fruits and flowers.

Ethics. All work was in accordance to local customs and the guidelines of the American Society of Mammalogists for the use of wild animals in research {Sikes, 2011 #29 573} and was carried out under approval from local authorities. The work in Zambia was approved by the Zambia Wildlife Authority (ZAWA 421902, 29/11/13, ZAWA 547649, 26/11/14).

Data accessibility. Data are available on the Movebank Data Repository https://doi.org/10.5441/001/1.k8n02jn8 [55].

Authors' contributions. M.C.C.-C., M.T.O., D.K.N.D. and J.F. conceived the project. J.F., M.W. and R.K. collected data. M.C.C.-C. performed the analyses. M.C.C.-C., M.T.O. and D.K.N.D. wrote the manuscript, and all authors approved the final version of the manuscript.

Competing interests. The authors declare no competing interests.

Funding. This study was supported by the Max Planck Institute of Animal Behaviour, the Max Planck Society, and field work in Zambia 2014 was supported through funds to the Institute of Novel and Emerging Infectious Diseases (Prof. Dr Martin H. Groschup, Friedrich-Loeffler-Institute, Greifswald, Germany) from the Federal Foreign Office of Germany (ref. no. ZMVI6-FKZ2513AA0374).

Acknowledgements. We thank Richard Suu-Ire for help with logistics and permits in Ghana, Lackson Chama for help with permits in Zambia, Michael Abedi-Lartey for data collection in Ghana and Burkina Faso, and Frank Willems, Sebastian Stockmaier and Natalie Weber for help in the field in Zambia. We also thank Anne Scharf for help with behavioural classification, and Sarah Davidson and the Movebank team for data curation.

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
