## [Reviewer comments · Royal Society Open Science]

Review History

RSOS-200274.R0 (Original submission)

Review form: Reviewer 1 (Paul Racey)

Is the manuscript scientifically sound in its present form?

Yes

Are the interpretations and conclusions justified by the results?

Yes

Is the language acceptable?

Yes

Do you have any ethical concerns with this paper?

No

Have you any concerns about statistical analyses in this paper?

No

Recommendation?

Accept with minor revision (please list in comments)

Comments to the Author(s)

I have only one quibble and that is the use of the word 'Surprisingly' (P15 l 267) in relation to bats at Kasanka travelling shorter mean distances than those at the smaller Accra colony despite the huge difference in colony size. 'Surprisingly' is a value judgement.

It does not surprise me given the superabundance of food (water berry, wild loquat and red milkwood) that I have observed in the miombo woodland surrounding the Kasanka roost, as well as mangoes in the nearest villages.

Review form: Reviewer 2**Is the manuscript scientifically sound in its present form?**

No

Are the interpretations and conclusions justified by the results?

No

Is the language acceptable?

Yes

Do you have any ethical concerns with this paper?

Yes

Have you any concerns about statistical analyses in this paper?

No

Recommendation?

Major revision is needed (please make suggestions in comments)

Comments to the Author(s)

This study promises to offer very interesting insights in the dynamics of large bat colonies and the influence on colony size on foraging behaviour. Unfortunately, it seems to me, that the study design is not adequate for the assumption made. Or at least, the uncertainties that come with the design should have been critically questioned much more in the discussion.

In the following section, I tried to refer to my criticism in detail:

- How many nights was each individual tracked? Might be added to Table S1. According to line 196, "at least one complete night" but that seems few data. How much on average? This is also relevant information for the roost fidelity.

- One or two sentences on whether the colonies are mixed-sex, how the dynamics within the colonies (maternity roosts? mating?) are and why only males were tracked would have been nice in the introduction or methods part.

- Figures S1: A short explanation of the presented data would be beneficial, axes labeling in the lower two figures? Explain different colors in 1c. Which nr of clusters (1b) was chosen for 1c?

- Line 249-251: why did you choose the mentioned distributions? Short explanation would be helpful. In general, the methods section is quite short. It would be very helpful for somebody who is not that familiar with the used tests to expand this section a bit.

- Table 1: Sadly, I don't understand the superscript letters used as marker of significance between colony groups (ab, a, b, and c?) So are those the results of the Kruskal-Wallis test (see line 273)? So "c" seems to differ from those who are "a", "b" or "ab". But how similar or different are colonies with "ab", "a", and "b" from each other? I feel like there should be some sort of legend to explain this.

- Where are the results of turkey test?

- In the results section it is sometimes hard to understand which tests produced the presented results (as the methods part is quite short and with little explanations)
 - Table 2 is also hard to understand
 - Line 349: So that bat also never used the main roost?
 - Line 351: so those seven bats only used two different day roosts?
 - Line 386: this sentence is not logical. The first part is related to the fact, that the two colonies travel larger distances than smaller colonies. The second part refers to the observation, that both large colonies (despite drastic differences in colony sizes) seemed to reach a similar maximal distance. There should not be an "or".
 - Line 400-402, this was already mentioned in 393 wasn't it?
 - Line 402: I don't understand that logic as flowering season and high colony size occurred simultaneously. So how is it possible to attribute the long commuting distances to either one?
 - Line 412: I thought it might also be possible that, while the flowers are more attractive (and thus Accra-high bats travelled larger distances to feed on them), kiwi-high might have stuck to fruits to avoid the long distances and more importantly competition with the larger Accra-high colony? So the shorter distances in Kibi might not be due to the fact, that the Kibi colony is smaller, but because the Accra colony as a potential competitor is larger. Thus it might be less of an inter-colony issue but rather interdependencies between colonies?
 - Also, in general, it seems to be difficult to compare colonies with so many different parameters
 - o Accra-high differs in their feeding habits of nectar from any other colony
 - o The Ouagadougou colony was not monitored in spring but only June-September with a possible effect of decreasing food abundance (line 416)
 - o Kasanka is only present during wet season anyway.
- So in my opinion there are too many factors that might play a role (different climates, food preference, food abundance, other seasonal aspects e.g. mating season??) but differ between the datasets. So it is very hard to compare the different data sets. I would have preferred to maybe just compare two colonies (with different colony sizes but otherwise very similar) during the same timeframes and across several years. The shortcomings in the study design are also not sufficiently addressed in the discussion part. Instead the authors made many assumptions without taking different possibilities into account.

Decision letter (RSOS-200274.R0)

31-Mar-2020

Dear Miss Calderon Capote,

The editors assigned to your paper ("Foraging movements are density-independent among straw-colored fruit bats") have now received comments from reviewers. We would like you to revise your paper in accordance with the referee and Associate Editor suggestions which can be found below (not including confidential reports to the Editor). Please note this decision does not guarantee eventual acceptance.

Please submit a copy of your revised paper before 23-Apr-2020. Please note that the revision deadline will expire at 00.00am on this date. If we do not hear from you within this time then it will be assumed that the paper has been withdrawn. In exceptional circumstances, extensions may be possible if agreed with the Editorial Office in advance. We do not allow multiple rounds of revision so we urge you to make every effort to fully address all of the comments at this stage. If deemed necessary by the Editors, your manuscript will be sent back to one or more of the original reviewers for assessment. If the original reviewers are not available, we may invite new reviewers.

To revise your manuscript, log into <http://mc.manuscriptcentral.com/rsos> and enter your

Author Centre, where you will find your manuscript title listed under "Manuscripts with Decisions." Under "Actions," click on "Create a Revision." Your manuscript number has been appended to denote a revision. Revise your manuscript and upload a new version through your Author Centre.

- Data accessibility

If you wish to submit your supporting data or code to Dryad (<http://datadryad.org/>), or modify your current submission to dryad, please use the following link:
<http://datadryad.org/submit?journalID=RSOS&manu=RSOS-200274>

- Competing interests

- Authors' contributions

- Acknowledgements

- Funding statement

Kind regards,

Andrew Dunn

on behalf of Dr Punidan Jeyasingh (Associate Editor) and Pete Smith (Subject Editor)

Associate Editor's comments (Dr Punidan Jeyasingh):

This manuscript was reviewed by two experts. While one was highly supportive, the other raises several questions. I felt the reviews were fair and constructive. I think the authors can allay the critical reviewer concerns via a revision. With much gratitude to the expert reviewers, I invite the authors to make these adjustments and submit a fresh version.

Comments to Author:

Reviewers' Comments to Author:

Reviewer: 1

Comments to the Author(s)

I have only one quibble and that is the use of the word 'Surprisingly' (P15 l 267) in relation to bats at Kasanka travelling shorter mean distances than those at the smaller Accra colony despite the huge difference in colony size. 'Surprisingly' is a value judgement.

It does not surprise me given the superabundance of food (water berry, wild loquat and red milkwood) that I have observed in the miombo woodland surrounding the Kasanka roost, as well as mangoes in the nearest villages.

Reviewer: 2

Comments to the Author(s)

This study promises to offer very interesting insights in the dynamics of large bat colonies and the influence on colony size on foraging behaviour. Unfortunately, it seems to me, that the study design is not adequate for the assumption made. Or at least, the uncertainties that come with the design should have been critically questioned much more in the discussion.

In the following section, I tried to refer to my criticism in detail:

- How many nights was each individual tracked? Might be added to Table S1. According to line 196, "at least one complete night" but that seems few data. How much on average? This is also relevant information for the roost fidelity.

- One or two sentences on whether the colonies are mixed-sex, how the dynamics within the colonies (maternity roosts? mating?) are and why only males were tracked would have been nice in the introduction or methods part.

- Figures S1: A short explanation of the presented data would be beneficial, axes labeling in the lower two figures? Explain different colors in 1c. Which nr of clusters (1b) was chosen for 1c?

- Line 249-251: why did you choose the mentioned distributions? Short explanation would be helpful. In general, the methods section is quite short. It would be very helpful for somebody who is not that familiar with the used tests to expand this section a bit.
 - Table 1: Sadly, I don't understand the superscript letters used as marker of significance between colony groups (ab, a, b, and c?) So are those the results of the Kruskal-Wallis test (see line 273)? So "c" seems to differ from those who are "a", "b" or "ab". But how similar or different are colonies with "ab", "a", and "b" from each other? I feel like there should be some sort of legend to explain this.
 - Where are the results of turkey test?
 - In the results section it is sometimes hard to understand which tests produced the presented results (as the methods part is quite short and with little explanations)
 - Table 2 is also hard to understand
 - Line 349: So that bat also never used the main roost?
 - Line 351: so those seven bats only used two different day roosts?
 - Line 386: this sentence is not logical. The first part is related to the fact, that the two colonies travel larger distances than smaller colonies. The second part refers to the observation, that both large colonies (despite drastic differences in colony sizes) seemed to reach a similar maximal distance. There should not be an "or".
 - Line 400-402, this was already mentioned in 393 wasn't it?
 - Line 402: I don't understand that logic as flowering season and high colony size occurred simultaneously. So how is it possible to attribute the long commuting distances to either one?
 - Line 412: I thought it might also be possible that, while the flowers are more attractive (and thus Accra-high bats travelled larger distances to feed on them), kiwi-high might have stuck to fruits to avoid the long distances and more importantly competition with the larger Accra-high colony? So the shorter distances in Kibi might not be due to the fact, that the Kibi colony is smaller, but because the Accra colony as a potential competitor is larger. Thus it might be less of an inter-colony issue but rather interdependencies between colonies?
 - Also, in general, it seems to be difficult to compare colonies with so many different parameters
 - o Accra- high differs in their feeding habits of nectar from any other colony
 - o The Ouagadougou colony was not monitored in spring but only June-September with a possible effect of decreasing food abundance (line 416)
 - o Kasanka is only present during wet season anyway.
- So in my opinion there are too many factors that might play a role (different climates, food preference, food abundance, other seasonal aspects e.g. mating season??) but differ between the datasets. So it is very hard to compare the different data sets. I would have preferred to maybe just compare two colonies (with different colony sizes but otherwise very similar) during the same timeframes and across several years. The shortcomings in the study design are also not sufficiently addressed in the discussion part. Instead the authors made many assumptions without taking different possibilities into account.

Author's Response to Decision Letter for (RSOS-200274.R0)

See Appendix A.

Decision letter (RSOS-200274.R1)

Dear Miss Calderon Capote,

It is a pleasure to accept your manuscript entitled "Foraging movements are density-independent among straw-colored fruit bats" in its current form for publication in Royal Society Open Science. The comments of the reviewer(s) who reviewed your manuscript are included at the foot of this letter.

on behalf of Dr Punidan Jeyasingh (Associate Editor) and Pete Smith (Subject Editor)
openscience@royalsociety.org

Associate Editor Comments to Author (Dr Punidan Jeyasingh):
Comments to the Author:

I thank the authors for thoroughly addressing reviewer comments. I am happy to recommend the manuscript for publication.

Appendix A

Dear Dr. Dunn

We thank you, the associate editors, especially Dr. Jeyasingh, and the two reviewers very much for the positive assessment of our manuscript. The comments are thoughtful and will help to make our paper clearer for the reader. Thus, we have incorporated them as suggested almost everywhere. Please find our detailed response below in red, preceded by ">" and with the new line numbers indicated.

In addition, we have also made **some changes which are marked in the text and we will list here:**

- Adjusted the methods modifying the subtitles inside this section, and included a section named activity budgets (line 240 – 243), to be congruent with the sequence followed in the results.
- We adjusted the heading in Figure 2.
- We adjusted the table 1 and show only 1 digit in the decimals when possible, except for the r square statistic.
- We adjusted some values in the results of the behavioral activity budget section which are marked with tracked changes.
- We remade table 2, rounding decimals of all values to one digit when possible.
- In the results section in the subheading: Day roost fidelity, we included a Kruskal-Wallis and a Dunn Test to test differences in the fidelity proportion across colonies (lines 273-274).
- We adjusted Figure S4: Proportion of day roost fidelity, based on previous mentioned test results.
- We remade Table S2 using only one decimal digit in all values.
- We remade Table S3 using only one decimal digit in all values, when possible.

We hope you will find the new version to your satisfaction and look forward to hearing from you.

Camila Calderon and co-authors

> we have added an ethics section

- Data accessibility

It is a condition of publication that all supporting data are made available either as supplementary information or preferably in a suitable permanent repository. The data accessibility section should state where the article's supporting data can be accessed. This section should also include details, where possible of where to access other

relevant research materials such as statistical tools, protocols, software etc can be accessed. If the data have been deposited in an external repository this section should list the database, accession number and link to the DOI for all data from the article that have been made publicly available. Data sets that have been deposited in an external repository and have a DOI should also be appropriately cited in the manuscript and included in the reference list.

<http://datadryad.org/submit?journalID=RSOS&manu=RSOS-200274>

> our data are available on the data repository Movebank - the DOI is given at the end of the manuscript (DOI: 10.5441/001/1.k8n02jn8)

- Competing interests

> we have no competing interests this is indicated at the end of the manuscript

- Authors' contributions

> author contributions are included

- Acknowledgements

> included as indicated

- Funding statement

> funding is reported at the end of the manuscript

on behalf of Dr Punidan Jeyasingh (Associate Editor) and Pete Smith (Subject Editor)
openscience@royalsociety.org

Associate Editor's comments (Dr Punidan Jeyasingh):

This manuscript was reviewed by two experts. While one was highly supportive, the other raises several questions. I felt the reviews were fair and constructive. I think the authors can allay the critical reviewer concerns via a revision. With much gratitude to the expert reviewers, I invite the authors to make these adjustments and submit a fresh version.

> dear Dr. Jeyasingh, thank you for handling this paper and we gladly comply. Please find our detailed answers to the reviewers below.

Comments to Author:

Reviewers' Comments to Author:

Reviewer: 1

Comments to the Author(s)

I have only one quibble and that is the use of the word 'Surprisingly' (P15 | 267) in relation to bats at Kasanka travelling shorter mean distances than those at the smaller Accra colony despite the huge difference in colony size. 'Surprisingly' is a value judgement.

It does not surprise me given the superabundance of food (water berry, wild loquat and red milkwood) that I have observed in the miombo woodland surrounding the Kasanka roost, as well as mangoes in the nearest villages.

> thank you for the positive assessment! We have deleted the word "surprisingly" of course, even though we really were surprised, given the density of resources we have seen at other sites. But obviously the bats agree with you and we stand corrected. Gladly.

Reviewer: 2

Comments to the Author(s)

This study promises to offer very interesting insights in the dynamics of large bat colonies and the influence on colony size on foraging behaviour. Unfortunately, it seems to me, that the study design is not adequate for the assumption made. Or at least, the uncertainties that come with the design should have been critically questioned much more in the discussion.

In the following section, I tried to refer to my criticism in detail:

> We are sorry you were so unhappy with the study design and hope we address your concerns properly in the new version and below. Thank you for your time and effort.

Reviewer 2

- How many nights was each individual tracked? Might be added to Table S1. According to line 196, "at least one complete night" but that seems few data. How much on average? This is also relevant information for the roost fidelity.

> We have now added the mean number of tracked nights to the text (Line 197-198). The number of nights per bat continues to be listed in Table S2.

- One or two sentences on whether the colonies are mixed-sex, how the dynamics within the colonies (maternity roosts? mating?) are and why only males were tracked would have been nice in the introduction or methods part.

> Thank you for this comment. For a reader who is more familiar with temperate zone bats this may indeed not be clear. *E. helvum* live in mixed-sex colonies year-round, we have added this to the general description of the colonies (Line 113)

> We now explain why females were not tracked on lines 177-178. These were early generation GPS loggers and they were too heavy for the smaller females.

- Figures S1: A short explanation of the presented data would be beneficial, axes labeling in the lower two figures? Explain different colors in 1c. Which nr of clusters (1b) was chosen for 1c?

> We have now added a more detailed explanation in the heading of Figure S1b-c. Also we changed the x and y axis in Figure S1c, which is referring to the coordinates in the UTM coordinate system.

- Line 249-251: why did you choose the mentioned distributions? Short explanation would be helpful. In general, the methods section is quite short. It would be very helpful for somebody who is not that familiar with the used tests to expand this section a bit.

> We now briefly explain the selection of the chosen distributions for the glmm models (lines 259-261).

- Table 1: Sadly, I don't understand the superscript letters used as marker of significance between colony groups (ab, a, b, and c?) So are those the results of the Kruskal-Wallis test (see line 273)? So "c" seems to differ from those who are "a", "b" or "ab". But how similar or different are colonies with "ab", "a", and "b" from each other? I feel like there should be some sort of legend to explain this.

> First, we have replaced the word "group" with "colony group" in the context of this analysis in the text and table as it was confusing. The superscripts refer to the results of a Tukey test (multiple comparison). The colonies sharing letters are not different from each other. For example, colonies with a, and ab do not differ significantly. And neither do colonies with ab and b. But a colony with a and a colony with b would differ from each other.

- Where are the results of turkey test?

> The results are summarized in Table 1 (see also answer above)

- In the results section it is sometimes hard to understand which tests produced the presented results (as the methods part is quite short and with little explanations)

> We have now restructured the analysis part in the methods section. We explain the analyses performed as it appears respectively in the results section. We also make clear in the results which analysis was used in each case. Additionally, we added the multiple comparison of the Dunn's test to Figure 3, hoping this makes these results clearer. Furthermore, we added a Kruskal-Wallis test to check differences across colonies in the fidelity proportion showed in Figure S4 (lines 273-274) and included the results of the Dunn test in the same Figure S4.

- Table 2 is also hard to understand

> We hope that between the changes explained above and the extended Table header, this is clearer now.

- Line 349: So that bat also never used the main roost?

> This bat whose track is shown in Figure S4c (bat number 1617) was away from the roost all tracked days. We make this also clear in the text (lines 369-370).

- Line 351: so those seven bats only used two different day roosts?

> We have double checked our results and realized that six not seven bats had two different day roosts and only one bat had four different roost (lines 373-375). Along

these lines, we have performed a Kruskal-Wallis analysis to see difference between the colonies (line 362-364).

- Line 386: this sentence is not logical. The first part is related to the fact, that the two colonies travel larger distances than smaller colonies. The second part refers to the observation, that both large colonies (despite drastic differences in colony sizes) seemed to reach a similar maximal distance. There should not be an “or”.

> You are right, thank you this observation. We have rewritten this sentence and hope that it is now clearer for the readers (line 409).

- Line 400-402, this was already mentioned in 393 wasn't it?

> You are right, we removed it.

- Line 402: I don't understand that logic as flowering season and high colony size occurred simultaneously. So how is it possible to attribute the long commuting distances to either one?

> It is not possible to know this, therefore our comparison with the Kasanka colony.

Line 412: I thought it might also be possible that, while the flowers are more attractive (and thus Accra-high bats travelled larger distances to feed on them), kiwi-high might have stuck to fruits to avoid the long distances and more importantly competition with the larger Accra-high colony? So the shorter distances in Kibi might not be due to the fact, that the Kibi colony is smaller, but because the Accra colony as a potential competitor is larger. Thus it might be less of an inter-colony issue but rather interdependencies between colonies?

Also, in general, it seems to be difficult to compare colonies with so many different parameters

- o Accra- high differs in their feeding habits of nectar from any other colony
- o The Ouagadougou colony was not monitored in spring but only June-September with a possible effect of decreasing food abundance (line 416)

- o Kasanka is only present during wet season anyway.

> You are correct with all of these points. For example, as indicated in the methods section the Ouagadougou colony is seasonal, it is present only from February to September, and absent the rest of the year. Thus, we do not want to overinterpret the results and focus on the parameter we are interested in, colony size. There are many other reasons why the colonies could and should differ, but our baseline assumption

was that such an enormous difference in size should have an effect regardless. This assumption was wrong and future studies will have to address alternative explanations (and should).

So in my opinion there are too many factors that might play a role (different climates, food preference, food abundance, other seasonal aspects e.g. mating season??) but differ between the datasets. So it is very hard to compare the different data sets. I would have preferred to maybe just compare two colonies (with different colony sizes but otherwise very similar) during the same timeframes and across several years. The shortcomings in the study design are also not sufficiently addressed in the discussion part. Instead the authors made many assumptions without taking different possibilities into account.

> While you are correct that there are many factors that may play a role, the bats have to eat during all times of the year. And as (predominant) central place foragers they are likely to compete – we think that is a valid assumption.

We are not sure what you mean by the many other assumptions we make. We feel like we mainly made the one: colony size should be linked to competition and thus foraging distance at these scales of magnitude. As you can infer from the methods we compared one new dataset (Kasanka) with previously published data, and while we agree that is not an ideal study design, we think it is valid to use these existing datasets, especially in such poorly known, but important species, where on-the-ground studies are logistically and financially difficult. Sure, we do not learn as much as we would have from a study where we went to several colonies, assessed resources and other parameters, but we do learn something in our humble opinion. For a more direct comparison within a colony, but between seasons, you could look at Fahr et al. 2014. The differences we found there actually triggered the current analysis.

We have added to the discussion to make this clearer.